# Son Preference and the Reproductive Behavior of Rural-Urban Migrant Women of Childbearing Age in China: Empirical Evidence from a Cross-Sectional Data

**DOI:** 10.3390/ijerph17093221

**Published:** 2020-05-06

**Authors:** Xiaojie Wang, Wenjie Nie, Pengcheng Liu

**Affiliations:** 1School of Management, Ocean University of China, Qingdao 266100, China; xiaojie_nk@126.com (X.W.); nie_wen_jie@163.com (W.N.); 2School of Economics, Qingdao University, Qingdao 266100, China

**Keywords:** son preference, reproductive behavior, migrant women, subsequent parity

## Abstract

Son preference has been shown to influence the childbearing behavior of women, especially in China. Existing research has largely focused on this issue using cross-sectional data of urban or rural populations in China, while evidence from the rural-urban migrant women is relatively limited. Based on the data of China Migrants Dynamic Survey in 2015, we used logistic regression models to explore the relationship of son preference and reproductive behavior of rural-urban migrant women in China. The results show that the son preference of migrant women is still strong, which leads women with only daughters to have significantly higher possibility of having another child and results in a higher imbalance in the sex ratio with higher parity. Migrant women giving birth to a son is a protective factor against having a second child compared to women whose first child was a girl. Similarly, the effects of the gender of the previous child on women’s progression from having two to three children showed the same result that is consistent with a preference for sons. These findings have implications for future public strategies to mitigate the son preference among migrant women and the imbalance in the sex ratio at birth.

## 1. Introduction

A strong preference for sons over daughters is common in East and Southeast Asia [1,2], notably in China. Son preference is often thought to be an important cause of imbalance in the sex ratio at birth [3,4,5]. According to the Global Gender Gap Report 2018, China ranked dead last among 149 countries in terms of “sex ratio at birth”. This is the result of many factors, such as the Confucian cultural tradition, the socioeconomic system, and gender ideology. In the context of traditional patriarchal, patrilineal and patriarchal systems, sons are considered to have unique value, as they inherit the family name and property and represent an economic value premium to the family and parents [6,7,8,9,10].

Significantly different from that in other countries, the family planning policy in China is one of the limiting factors affecting fertility practices in the country [11,12,13,14]. Although the policy has been loosened by the shift from allowing one child to allowing two children for all couples, a clear limitation on the number of children still exists [15]. When there is a difference between the number of children allowed by the family planning policy and the number of children expected by the family, for individuals with a strong preference for boys, there is an incentive to conduct gender selection under the policy conditions or ignore the normal fertility limitation to have more children until a son is born. It is worth noting that the occurrence of the above two cases is strongly related to the individual’s household registration status [16].

Another special policy in China, the household registration policy, also has a significant impact on women’s reproductive behavior. The dual structure in urban and rural areas in China leads to significant differences in economic development, children-bearing concepts, and policy supervision [17,18,19]. The fertility behavior of urban residents is heavily regulated, and the opportunity cost of violating policies is extremely high, so the number of births remains low [20,21,22], and it is rare to maintain son preference through having more children. However, in addition to urban residents, there are large groups of rural-to-urban migrants in cities. Rural-to-urban migrants are people who have resided in an urban destination for at least 6 months and do not have local household registration [23]. The migrant population is too large to be ignored, accounting for approximately one-sixth of the total population of China [24]. According to the National Bureau of Statistics, the number of migrants reached 241 million in 2018, and half of them were migrant women. A considerable proportion of these women were of child-bearing age [25]. Although they make a living in the city, traditional concepts such as having a son to carry on the family name, raising children to provide for parents in old age and promoting the family status of mothers by giving birth to a boy are deeply entrenched in their minds, which motivates them to have more children and give birth to at least one boy [26]. Furthermore, the opportunity cost of violating policies is lower for migrant women than for urban women, so the fertility behavior of continuing to have children until a son is born is more likely to be observed in this group. However, scholars have pointed out that the well-bear and well-rear concept of urban residents may have a demonstratable effect on migrants and gradually change their fertility conception [19,27,28]. Studies have shown that migrants generally have lower fertility rates than rural residents and higher fertility rates than urban residents [27,29].

To a large extent, the birth conception and behavior of migrant women have a profound impact on the population changes in the places of domicile and migration, which will further affect the implementation of China’s family planning policy, the overall fertility level and the changes in the gender structure [30]. Moreover, due to the high mobility of this type of population, the government has difficulties conducting cross-regional fertility regulations. Therefore, the constraints of the family planning policy on migrants are greatly decreased, so fertility intention is more likely to be implemented; that is, there may be a high degree of consistency between the migrating population’s reproductive preference for sons and future fertility behaviors [31]. In summary, under the special policy background of China, choosing migrant women as the research object has certain practical significance for studying the relationship between the son preference and reproductive behavior of migrant women of childbearing age, predicting future population development trends and reversing the gender imbalance.

Studies on women’s reproductive behavior in the existing literature focus on family reproductive intention, family planning, number of sons and contraceptive usage [8,12,14,32,33]. For example, prospective analysis is performed by directly asking questions such as “How many children do you expect to have?”, “Would you like to have another child?” and other questions at the level of fertility intention to conduct a prospective analysis of fertility behavior. However, whether fertility intention can be transformed into actual fertility behavior is still limited by a variety of factors, such as national policies, family economic conditions and physical conditions of the couple, and the internal deviation between fertility intention and behavior cannot be effectively estimated. In particular, there is also a large deviation between son preference in terms of fertility intention and gender selection in actual reproductive behavior. Therefore, the direct investigation of fertility behavior can better reflect the issue of son preference.

The influence of son preference on the number of children and fertility behavior remains a controversial topic. Some studies suggest that a strong preference for sons will lead to the increases in fertility level desired by the state [34,35]. Others have shown that using sex-selective technologies will lead to a reduction in the number of children and thus a lower fertility level. In terms of gender composition, although evidence from India, South Korea, and Vietnam suggests that the preference for sons is closely related to family fertility behavior, couples with only girls are likely to have more children than families with only boys or gender-balanced families [2,8,36,37,38,39]. However, in the context of China’s special policy, relatively few studies have focused on rural-urban migrant women to investigate the individual characteristics and social factors that may affect fertility behavior. The complex relationship between son preference and actual fertility behavior has not been effectively tested, and it remains to be explored whether the migrant experience can influence the son preference and fertility behavior of immigrant women. In addition, this article can help overcome the adverse bias between fertility intention and fertility behavior in previous studies, and the retrospective research method adopted largely makes the research conclusions more reliable and convincing.

Regarding the research on the son preferences of migrant women, this paper conducts a retrospective study based on the 2015 national monitoring survey data of the China Migrants Dynamic Survey (CMDS) and focuses on two aspects. The first is the pursuit of the subsequent parity progression. The second is the relationship between the sex composition of existing children and subsequent childbearing behavior. That is, in the context of China’s special fertility policy, what are the factors that influence migrant women to have a higher number of children? If the gender composition of existing children fails to satisfy migrant women’s son preference, will the women violate national policies and have more children? As the number of children increases, will the likelihood of having boys be increased through prenatal sex identification technology? Having more children can reduce the quality of childcare, while choosing to have boys following prenatal sex identification may cause severe gender imbalances, both of which pose challenges to improving the quality of the population and ensuring the gender balance in China.

## 2. Materials and Methods

### 2.1. Data Source

The data in this paper are derived from the China Migrants Dynamic Survey in 2015 (CMDS2015). This is an annual large-scale national migrant population sampling survey initiated by the National Health Commission of the People’s Republic of China and coordinated by the China Population and Development Research Center. CMDS2015 covers 31 provinces (autonomous regions and municipalities) in mainland China. The stratified, multistage and proportional PPS method is adopted for sampling. The survey covers the basic information of the migrant individuals and their family members, the mobility range of the migrant women, employment and social security, income and expenditure, residence, basic public health services, management of marriage and family planning services, children’s mobility and education, etc. The survey questionnaire was presented through direct interviews by investigators who had been trained uniformly and reviewed by professional instructors. After the questionnaire was completed, it was properly kept in each city (district) and randomly checked by a panel of experts. Finally, the quality of the submitted questionnaire was monitored by logical verification and telephone return visits. The CMDS2015 was characterized by the full coverage and representativeness of the migrant population, and the questionnaire has good authenticity and reliability.

### 2.2. Study Design and Participants

Cross-sectional data of rural-to-urban migrant women with one child were derived from CMDS2015 to explore the complex relationship of son preference and reproductive behavior. Given the research question of our study, we limited our sample to migrant women of childbearing age. Specifically, the sample selection criteria in this article are being married (first marriage or remarriage), having had at least one child, being aged 15–49 years old, having rural household registration, and being a non-local resident. Following the exclusion of non-qualifying individuals and invalid data samples resulting from refusal to answer, the number of valid samples was 36,182.

The dependent variable in this study is the fertility behavior of migrant women, that is, whether each respondent progressed from having one, two and three children to having two, three and four children, respectively. The independent variable is the gender of existing children. The control variables involved in this study are grouped into two categories. The first category elicited sociodemographic characteristics of the participants, including age, ethnic group, marriage duration, individual education level, spouse’s education level, employment, mobility range and residence intention. The second category assessed the participants’ fertility information based on the following items: the number of existing children and the birthplace of the first child. The proportion of boys being born should be around 51 percent without son preference, and if the proportion in our data is higher than this ratio, it means that the boy preference exists. Since the outcome variable of this study is a binary variable, we used logistics regression analysis to explore the effects of motivation and influencing factors, especially the sex composition of current children, on the subsequent childbearing behavior of migrant women.

### 2.3. Statistical Analysis

A descriptive analysis was used to describe the characteristics of the participants and their reproductive behaviors. Variables that were significantly associated with fertility behavior by chi-square analysis were entered as independent variables in the binary logistic regression, including age, ethnic group, marriage duration, individual education level, spouse’s education level, gender of the first child, mobility range and residence intention. The factors which were significantly related to reproductive behavior were then included in the logistic model analysis. A single-factor regression analysis was used to explore the effect of the gender of existing children on having another child. A multivariable logistic regression model was built to identify the determinants of reproductive behaviors. Through these analyses, crude ORs and 95% confidence intervals (CIs) were estimated for the gender preference and fertility behavior of migrant women. Statistical significance was defined as *p*-values < 0.05. Data analysis was conducted using the STATA 15.0. (Stata, College Station, TX, USA).

## 3. Results

### 3.1. Sociodemographic Characteristics

Table 1 presents the descriptive statistics of key variables, showing that in the overall sample of 36,182, most of the respondents were between the ages of 25 and 44 (81.88%), and the majority were ethnic Han, while less than 10% were minorities. Generally, the respondents were not highly educated, with only 6% of migrant women having a bachelor’s degree or above and nearly 60% having a junior high school education (57.32%), while those who had no education or only primary education accounted for 2.25% and 16.4%, respectively. Most women had stable jobs (72.77%). Migrant women with one, two and three children accounted for 57.25%, 38.06%, and 4.18%, respectively, while less than 1% of the sample had more than three children. Most of the migrant women chose to have their first child in the hospital (89.36%), while 9.4% and 1.24% of migrant women chose to bear their first child at home or in a private clinic, respectively. Regarding the geographical range of migration, nearly half of them engaged in interprovincial migration (47.22%), 31.44% were intercity migrants, and 21.34% were intercounty migrants. A total of 85.03% of migrant women had the intention to live for a long time in their destinations.

### 3.2. Factors Affecting Childbearing Behavior

Variables that were significantly associated with fertility behavior by chi-square analysis were entered as independent variables in the binary logistic regression. We took fertility behavior for the second child as an example and conducted an analysis. As shown in Table 2, the frequency of having a second child for women whose first child was a girl (53.53%) was significantly higher than those whose first child was a boy (33.18%). Migrant women who have been married for more than 5 years have a much higher proportion of progressing to second births (48.10%) than women who have been married for less than 5 years (6.32%), which shows the increased marriage duration improved the chance of having a second child. There is a significant positive correlation between age and having a second child. With increasing age, the proportion of second children gradually increases. In contrast, with the improvement of women’s education level, the proportion of women choosing to have a second child gradually decreases from 73.13% to 13.17%. The education level of the husband shows a similar trend. Migrant women who have stable jobs and want to live in the destination cities have higher frequency of progressing to the next birth than women who do not work or want to stay.

### 3.3. Sex Composition and Subsequent Parity Progression

Table 3 shows the distribution of the sex composition of existing children. Generally, the proportion of migrant women who progressed from having one, two, and three children to having two, three, and four children, respectively, is gradually decreasing. That is, among the children with a total sample size of 53,530, first and second children represented 36,182 and 15,468, accounting for 67.59% and 28.90%, respectively, while the number of third and fourth children sharply reduced to 1697 (3.17%) and 183 (0.34%), respectively. In terms of sex composition, the sex ratio significantly increases with the birth of more children. The sex ratio of the second children (1.42) is higher than that of the first children (1.13). Although the number of third and fourth children is relatively small, the sex ratio reaches a staggering 1.81 and 2.27, respectively. All the above are significantly higher than the natural sex ratio (1.06).

Table 4 shows the effects of the sex composition of children at baseline on women’s parity progression from having one, two, and three children to having two, three, and four children, respectively. Considering that younger women may not have completed their childbearing progression [40], we further limited the sample to migrant women of childbearing age over 35, and the sample size narrowed to 20,433. It can be seen that in the transition from the first to the second children, 68.60% of those whose first child was a girl chose to have a second child, while only 43.02% of those whose first child was a boy chose to have a second child. This suggests that the gender of the first child is significantly associated with the probability of progressing to the next parity. In the transition to having a third child, among women whose children were both daughters, the proportion of those who had a third child was 32.16%, significantly higher than the proportion among women whose children were both sons (less than 10%). Among women with three children, relative to women with three sons, those with three daughters had nearly four times the frequency of progressing to the next parity. In addition, women who had a son and a daughter were the most likely to stop having children. Similarly, among families with three children, women with two sons and one daughter were most likely to stop having children. Our findings show patterns of association between the sex composition of existing children and subsequent childbearing behaviors that are consistent with a preference for sons. (All the above model test results are statistically significant).

Table 5 shows the results of the single-factor regression analysis of the gender of existing children on having another child. In the transition from the first to the second child, migrant women with one son is a protective factor against having a second child compared to women whose first child was a girl (OR: 0.43; 95% CI: 0.413–0.450). Among the migrant women with two children, those with one (OR: 0.19; 95% CI: 0.168–0.211) or two sons (OR: 0.21; 95% CI: 0.180–0.246) had a significantly reduced chance of progressing to the next parity compared to women with two daughters. Similarly, the effects of the gender of the previous child on women’s progression from having three to four children show the same result that are consistent with a preference for sons. In addition, the son preference effect is significantly enhanced with the birth of more girls.

### 3.4. The Effect of Relevant Factors on Subsequent Childbearing

Furthermore, we took fertility behavior for the second child as an example and conducted a multifactor analysis. The results are reported in Table 6. Binary logistic regression analysis was used to explore various factors influencing whether migrant women chose to have a second child. The results showed that ethnic group, education level, residence intention, mobility range and having a job had a significant influence on the fertility behavior of migrant women (*p* < 0.05). Women with one son is a protective factor against having a second child compared to women whose first child was a girl (OR = 0.35, *p* < 0.05). Compared with minority women, a smaller proportion of Han women chose to have a second child due to the restriction of family planning policy. Women over 25 were more likely to have a second child than women aged 24 or younger. Migrant women with age of more than 35 years old increased the chance of progressing to the next parity by more than three times compared to women less than 25 years old (OR = 3.75, *p* < 0.05) (OR = 3.53, *p* < 0.05). Migrant women with education is a protective factor; those with higher education levels reported lower possibility of having another child than those who had never gone to school. The marriage duration more than 5 years increased the chance of having a second child by more than seven times compared to a marriage duration less than 5 years (OR = 7.42, *p* < 0.05). Having a job and the education level of the spouse were negatively correlated with having another child. Women with residence intention were more likely than women without residence intention to have a second child (OR = 1.19, *p* < 0.05). In addition, those who chose to have their first child at home (OR = 1.89, *p* < 0.05) or in a private clinic (OR = 2.08, *p* < 0.05) increased the chance of having a second child than those who had their first child in the hospital.

## 4. Discussion

China’s migrant population is large, and there is a trend of further growth in the future [23,24,41]. The fertility concepts of migrant women, to some extent, aggravate the complexity of the fertility level and gender structure of cities and even the whole of China, which will have a profound impact on future marriage, family structure, family planning policy and social development [19,27]. Based on the data of CMDS2015, this paper puts forward new views on the son preference and reproductive behavior of rural-urban migrant women of childbearing age in China.

The fertility concepts of migrant women tend to be markedly different from those of their rural or urban counterparts, which are influential in women’s preference regarding childbearing. Although the emerging literature indicated that the modern outlook of fertility in urban areas might reduce the parental preference for sons [27,42], some scholars argued that the son preference remained strong among migrants [7]. Our analysis confirmed the latter view: migrants still have a strong son preference in the context of China’s special policy, which makes having only daughters less desirable and results in a serious imbalance in the sex ratio.

Our findings showed the patterns of association between the sex composition of existing children and subsequent childbearing behaviors that are consistent with the preference for sons, which leads women with only daughters to have significantly higher intentions of having another child and results in a higher imbalance in the sex ratio with higher parity. To elaborate, our study found that migrant women whose first child was a girl were more likely to have another child compare with women whose first child was a boy. The same is true for the second and third children. It can be said that although the migrant women live in the city, the traditional concept of son preference had not changed significantly, which motivated them to have more children and give birth to at least one boy [7]. In addition, this study found that migrant women with both sons and daughters had the lowest odds of having another child. This also indicates that migrant women have a significant preference for sons. However, if the son preference is satisfied, the sex composition of children including both boys and girls is ideal for migrant women.

Socioeconomic characteristics can explain why some migrant women were more inclined to have more than one child. The results showed that with the increase in age, the likelihood of having another child gradually increased. A possible reason is that with the increase in age, individuals obtain a certain amount of savings and a stable work and residence environment that ease the pressure of raising another child, on the basis of which son preference can be achieved. Education level was found to have a significant effect on fertility. For better-educated migrants, the likelihood of having another child decreased greatly, and the evidence of a son preference declined. On the one hand, the increase in education improves migrant women’s awareness of and compliance with national policies. On the other hand, highly educated women pay more attention to the quality of childrearing than to the quantity of children [27]. Therefore, encouraging women to receive higher education is an important measure to alleviate China’s sex ratio imbalance.

In addition, the sex ratio of first children is slightly higher than the sex ratio under natural conditions (1.04–1.06) [7], indicating that the motivation for the sex selection of first children is not significant. The sex ratio of the second children (1.42) was significantly higher than that of the first children (1.13). Moreover, under family planning policy restrictions, the gender imbalance among third and fourth children worsened further, and the fourth children were more than twice as likely to be boys than girls. This result indicated an obvious tendency for gender selection and a motivation to ignore family planning restrictions and have more children in the case that the gender preference for sons was not satisfied. Therefore, migrant women who gave birth to the first child at home or in a private clinic had significantly higher odds of having a second child than those who gave birth to the first child in the hospital because the former can avoid inspection by family planning authorities.

Our study makes a possible contribution to the literature regarding the effect that son preference and the sex composition of existing children has on the reproductive behavior of women of childbearing age in the rural-urban migrant population. First, most studies of son preference focus on rural areas or locally representative samples, and there are relatively few studies on migrant women, a special group of women who differ from women in rural and urban areas. This paper expands the research scope of relevant fields. Second, most studies tend to explore fertility behaviors through fertility intentions, which are measured at a particular time point and could be unreliable because intentions are fluid. With data and retrospective birth histories, we can use logistic regression analysis to estimate the parity progression among migrant women conditional on sex composition.

The limitations of the study also should be highlighted. First, our study is based on a cross-sectional study that focuses on the number and gender of existing children, which makes it difficult to analyze the dynamics of female fertility. Women’s reproductive behavior could be tracked using more effective methods. If large-scale data covering migrant women can be collected in future studies, through a longitudinal study, a more comprehensive understanding of the gender preference and reproductive behavior will become possible. Second, married women rarely migrate independently. In fact, they often migrant with their husbands [43]. Under the gender norms of a patriarchal society, immigrant women may have little influence on family decisions, such as those involving the optimal number of births. Future research could also examine gender preference of husbands in the rural-urban migrant population. Third, there are various differences in the inflow and outflow areas, and the lack of regional cultural surveys have limited the conduct of our research. This requires more detailed data, which are not typically covered by the China Migrants Dynamic Survey. Thus, future research and evaluation studies with a special focus on these issues will be critical to push research in such a direction.

## 5. Conclusions

China’s long-term gender imbalance deserves much attention. Gender imbalance is closely related to son preference. Therefore, the change in the fertility concept of migrant women plays an important role in reversing the gender imbalance. However, under the existing household registration system, migrant women have not been truly integrated into urban society, which makes it difficult for them to adapt to both the urban and the rural cultures. They have become a marginal group that is separated from rural and urban areas but also closely related to these areas, which complicates the transformation of their fertility concept. Therefore, how to carry out selective intervention through public policies, especially aiming to reverse the son preference of migrant women, is a problem worthy of attention in the future.

## Figures and Tables

**Table 1 ijerph-17-03221-t001:** Sociodemographic characteristics of the study participants (*N* = 36,182).

Variables	*N*	%
Age of women (years)		
≤24	657	1.82
25–34	15,092	41.71
35–44	14,535	40.17
45–49	5898	16.3
Marriage duration (years)		
<5	4633	12.8
≥5	31,549	87.2
Ethnic group		
Minority	2744	7.58
Han	33,438	92.42
Highest education level		
No education	815	2.25
Primary school	5934	16.4
Junior high school	20,738	57.32
High school/Secondary school	6493	17.95
University and above	2202	6.09
Highest education level of husband		
Junior high school and below	18,669	79.92
High school/Secondary school and above	4690	20.08
Have a job		
No	9853	27.23
Yes	26,329	72.77
The number of existing children		
1	20,714	57.25
2	13,771	38.06
3	1514	4.18
≥4	182	0.50
Gender of the first child		
Female	17,021	47.04
Male	19,161	52.96
Where the first child was born		
Hospital	32,331	89.36
Private clinic	449	1.24
At home	3402	9.4
Scope of migration		
Trans-provincial	17,086	47.22
Intercity	11,374	31.44
Across the county	7722	21.34
Settlement intention		
No	5415	14.97
Yes	30,767	85.03

**Table 2 ijerph-17-03221-t002:** Distribution of second-child fertility by sociodemographic variables (*N* = 36,182).

Variables	Whether a Second Child Was Born					
	No		Yes		*N*	X^^2^ (*p*)
Gender of the first child	N	%	N	%		
Female	7910	46.47	9111	53.53	17,021	1.5 × 10^3^ (0.0000)
Male	12,804	66.82	6357	33.18	19,161	
Age of women (years)						
≤24	608	92.54	49	7.46	657	3.0 × 10^3^ (0.000)
25–34	10,890	72.16	4202	27.84	15,092	
35–44	6804	46.81	7731	53.19	14,535	
45–49	2412	40.90	3486	59.10	5898	
Marriage duration (years)						
<5	4340	93.68	293	6.32	4633	2.9 × 10^3^ (0.000)
≥5	16,374	51.90	15,175	48.10	31,549	
Ethnic group						
Minority	1343	48.94	1401	51.06	2744	83.7028 (0.000)
Han	19,371	57.93	14,067	42.07	33,438	
Highest education level						
No education	219	26.87	596	73.13	815	2.8 × 10^3^ (0.000)
Primary school	2123	35.78	3811	64.22	5934	
Junior high school	11,770	56.76	8968	43.24	20,738	
High school/Secondary school	4690	72.23	1803	27.77	6493	
University and above	1912	86.83	290	13.17	2202	
Highest education level of husband						
Junior high school and below	13,257	51.42	12,525	48.58	25,783	1.9 × 10^3^ (0.000)
High school/Secondary school and above	7457	71.70	2943	28.30	10,400	
Have a job						
No	5831	59.18	4022	40.82	9853	63.2967 (0.000)
Yes	14,883	56.53	11,446	43.47	26,329	
Where the first child was born						
Hospital	19,626	60.70	12,705	39.30	32,331	1.5 × 10^3^ (0.000)
Private clinic	150	33.41	299	66.59	449	
At home	938	27.57	2464	72.43	3402	
Scope of migration						
Trans-provincial	9115	53.35	7971	46.65	17,086	201.5062 (0.000)
Intercity	6896	60.63	4478	39.37	11,374	
Across the county	4703	60.90	3019	39.10	7722	
Settlement intention						
No	3223	59.52	2192	40.48	5415	13.4120 (0.000)
Yes	17,491	56.85	13,276	43.15	30,767	

**Table 3 ijerph-17-03221-t003:** Distribution of the sex composition of existing children.

	Female	%	Male	%	*N*	Sex Ratio
Gender of the first children	17,021	47.04%	19,161	52.96%	36,182	1.13
Gender of the second children	6404	41.40%	9064	58.60%	15,468	1.42
Gender of the third children	603	35.53%	1094	64.47%	1697	1.81
Gender of the fourth children	56	30.60%	127	69.40%	183	2.27
Total	24,084	44.99%	29,446	55.01%	53,530	1.22

**Table 4 ijerph-17-03221-t004:** Baseline sex composition on subsequent childbearing (*N* = 20,433).

Sex Composition of Previous Children at Baseline	Stopped Childbearing	%	Continued Childbearing	%	*N*	X^^2^ (*p*)
Gender of the first children						
0 male and 1 female	2979	31.40%	6509	68.60%	9488	1.3 × 10^3^ (0.000)
1 male and 0 female	6237	56.98%	4708	43.02%	10,945	
Gender of the first two children						
0 male and 2 female	1500	67.84%	711	32.16%	2211	977.7714 (0.000)
1 male and 1 female	6187	92.62%	493	7.38%	6680	
2 male and 0 female	2130	91.57%	196	8.43%	2326	
Gender of the first three children						
0 male and 3 female	94	59.12%	65	40.88%	159	145.3611 (0.000)
1 male and 2 female	704	91.91%	62	8.09%	766	
2 male and 1 female	364	92.39%	30	7.61%	394	
3 male and 0 female	72	88.89%	9	11.11%	81	

**Table 5 ijerph-17-03221-t005:** Single-factor regression analysis of the gender of the previous child on having another child.

Sex Composition of Previous Children at Baseline	OR	Coef	Std. Err.	Z	*p*	95% CI
gender of the first children						
0 male and 1 female	1					
1 male and 0 female	0.43	−0.84	0.01	−38.75	0.000	(0.413, 0.450)
gender of the first two children						
0 male and 2 female	1					
1 male and 1 female	0.19	−1.67	0.01	−28.75	0.000	(0.168, 0.211)
2 male and 0 female	0.21	−1.56	0.02	−19.64	0.000	(0.180, 0.246)
gender of the first three children						
0 male and 3 female	1					
1 male and 2 female	0.15	−1.88	0.03	−9.78	0.000	(0.104, 0.222)
2 male and 1 female	0.14	−1.94	0.03	−8.32	0.000	(0.091, 0.227)
3 male and 0 female	0.19	−1.67	0.07	−4.42	0.000	(0.089, 0.393)

**Table 6 ijerph-17-03221-t006:** Binary logistic regression analysis of factors influencing reproductive behavior.

Variables	Coef.	OR	Std. Err.	z	*p*	95% CI
Gender of the first children (Female as ref)	−1.05	0.35	0.01	−43.06	0.000	(0.334, 0.368)
Ethnic group (Minority as ref)	−0.24	0.79	0.04	−5.10	0.000	(0.719, 0.864)
Marriage duration (years) (<5 as ref)	2.00	7.42	0.49	30.11	0.000	(6.512, 8.453)
Age of women (years) (≤24 as ref)						
25–34	0.68	1.97	0.32	4.13	0.000	(1.429, 2.724)
35–44	1.32	3.75	0.62	7.98	0.000	(2.708, 5.128)
45–49	1.26	3.53	0.59	7.53	0.000	(2.541, 4.900)
Highest education level (No education as ref)						
Primary school	−0.23	0.80	0.07	−2.55	0.011	(0.668, 0.949)
Junior high school	−0.69	0.50	0.04	−7.76	0.000	(0.424, 0.599)
High school/Secondary school	−1.08	0.34	0.03	−11.19	0.000	(0.282, 0.411)
University and above	−1.61	0.20	0.02	−13.47	0.000	(0.158, 0.253)
Highest education level of husband (Junior high school and below as ref)	−0.16	0.85	0.02	−7.26	0.000	(0.820, 0.892)
Have a job (No as ref)	−0.20	0.82	0.02	−7.16	0.000	(0.775, 0.865)
Settlement intention (No as ref)	0.17	1.19	0.04	5.11	0.000	(1.113, 1.271)
Where the first child was born (Hospital as ref)						
Private clinic	0.64	1.89	0.20	5.93	0.000	(1.531, 2.330)
At home	0.73	2.08	0.09	16.44	0.000	(1.910, 2.275)
Scope of migration (Trans-provincial as ref)						
Intercity	−0.31	0.74	0.02	−10.98	0.000	(0.697, 0.778)
Across the county	−0.35	0.70	0.02	−11.26	0.000	(0.659, 0.746)
_cons	−1.28	0.28	0.05	−6.81	0.000	(0.193, 0.402)

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
