# Peer review of "Son Preference and the Reproductive Behavior of Rural-Urban Migrant Women of Childbearing Age in China: Empirical Evidence from a Cross-Sectional Data"

_ijerph, 2020, doi:10.3390/ijerph17093221_

Round 1

Reviewer 1 Report

This topic discussed in this manuscript is important. Below is my feedback:  

 1.      The abstract needs to be rewritten. The abstract is only written as a summary of the background. The authors have to rewrite it and make sure it incorporates a summary of the background, purpose of the study, the methodology, the findings and implications.

2.      The paper is well-structured. However, the authors have to add a section on limitations of the study

3.      Throughout the paper, there were some sweeping arguments by the authors which need to be referenced appropriately. For instance, on page 2, lines 6 and 7 need to be referenced appropriately. Again, on page 2 lines 7 and 8, you refer to scholars and studies but end up providing a single citation at the end of line 8. You should provide more than one references or citations.  

Generally, the paper is sound.

Author Response

We would like to thank you for the positive and constructive comments and suggestions. The corresponding modifications have been marked in red or in a modified version. Please see the attachment.

Reviewer 2 Report

The manuscript discusses extremely important cultural and gender issues, but some adjustments to the results and discussion are necessary. My suggestions are quoted in the file.

Author Response

(The authors gave the same response as above.)
